# Support Recovery in Sparse PCA with Incomplete Data

**Hanbyul Lee**
Department of Statistics
Purdue University
West Lafayette, IN 47906
lee3078@purdue.edu

**Qifan Song**
Department of Statistics
Purdue University
West Lafayette, IN 47906
qfsong@purdue.edu

**Jean Honorio**
Department of Computer Science
Purdue University
West Lafayette, IN 47906
jhonorio@purdue.edu

## Abstract

We study a practical algorithm for sparse principal component analysis (PCA) of incomplete and noisy data. Our algorithm is based on the semidefinite program (SDP) relaxation of the non-convex $l_1$-regularized PCA problem. We provide theoretical and experimental evidence that SDP enables us to exactly recover the true support of the sparse leading eigenvector of the unknown true matrix, despite only observing an incomplete (missing uniformly at random) and noisy version of it. We derive sufficient conditions for exact recovery, which involve matrix incoherence, the spectral gap between the largest and second-largest eigenvalues, the observation probability and the noise variance. We validate our theoretical results with incomplete synthetic data, and show encouraging and meaningful results on a gene expression dataset.

## 1 Introduction

Principal component analysis (PCA) is one of the most popular methods to reduce data dimension which is widely used in various applications including genetics, image processing, engineering, and many others. However, standard PCA is usually not preferred when principal components depend on only a small number of variables, because it provides dense vectors as a solution which degrades interpretability of the result. This can be worse especially in the high-dimensional setting where the solution of standard PCA is inconsistent as addressed in several works [Paul, 2007, Nadler, 2008, Johnstone and Lu, 2009]. To solve the inconsistency issue and improve interpretability, *sparse PCA* has been proposed, which enforces sparsity in the PCA solution so that dimension reduction and variable selection can be simultaneously performed. Theoretical and algorithmic researches on sparse PCA have been actively conducted over the past few years [Zou et al., 2006, Amini and Wainwright, 2008, Journée et al., 2010, Ma, 2013, Lei and Vu, 2015, Berk and Bertsimas, 2019, Richtárik et al., 2021].

In this paper, we consider a special situation where the data to which sparse PCA is applied are not completely observed, but partially missing. Missing data frequently occurs in a wide range of machine learning problems, where sparse PCA is no exception. There are various reasons and situations where data becomes incomplete, such as failures of hardware, high expenses of sampling, and preserving privacy. One concrete example is the analysis of single-cell RNA sequence (scRNA-seq) data [Park and Zhao, 2019], where the cells are divided into several distinct types which can be characterized with only a small number of genes among tens of thousands of genes. Sparse PCA can be effectively utilized here to reduce the dimension (from numerous cells to a few cell types) and to select a small number of genes that affect the reduced data. However, since scRNA-seq data usually have many missing values due to technical and sampling issues, the existing sparse PCA theory and method designed for fully observed data cannot be directly applied, and new methodology and theory are in demand.

36th Conference on Neural Information Processing Systems (NeurIPS 2022).

Despite the need for theoretical research and algorithmic development of sparse PCA for incomplete data, there have not been many studies yet. Lounici [2013] and Kundu et al. [2015] considered two different optimization objectives for sparse PCA on incomplete data, which impose $l_1$ regularization and $l_0$ constraint on the classic PCA loss function using a (bias-corrected) incomplete matrix, respectively. It was shown that the solution of each problem has a non-trivial error bound under certain conditions, but the optimization problems they considered are either nonconvex or NP-hard, and thus theoretical studies of computational feasible algorithms are still lacking. More recently, Park and Zhao [2019] proposed a computationally tractable two-step algorithm based on matrix factorization and completion, but its first step is an iterative algorithm that requires singular value decomposition in every iteration, which incurs a lot of cost in memory and time under a high-dimensional setting.

With this motivation, we suggest a computational friendly convex optimization problem via a semidefinite relaxation of the $l_1$ regularized PCA, to solve the sparse PCA on incomplete data. We note that very efficient scalable SDP solvers exist in practice [Yurtsever et al., 2021]. We assume that the unknown true matrix $\boldsymbol{M}^* \in \mathbb{R}^{d \times d}$ is symmetric and has a sparse leading eigenvector $\boldsymbol{u}_1$. Our goal is to exactly recover the support of this sparse leading eigenvector, i.e., to find the set $J$ correctly where $J = supp(\boldsymbol{u}_1) = \{i : u_{1,i} \neq 0\}$. Given a noisy observation $\boldsymbol{M}$ for the unknown true matrix $\boldsymbol{M}^*$, it is intuitive to consider imposing a regularization term on the PCA quadratic loss that aims to find the first principal component. When using the $l_1$ regularizer, the optimization problem can be written as:

$$\hat{\boldsymbol{x}} = \underset{\boldsymbol{x}^\top \boldsymbol{x} = 1}{\arg\max}\ \boldsymbol{x}^\top \boldsymbol{M} \boldsymbol{x} - \rho \|\boldsymbol{x}\|_1^2.$$

Hence, $J$ is estimated with $supp(\hat{\boldsymbol{x}})$. However, this intuitively appealing objective is nonconvex and very difficult to solve, so the following semidefinite relaxation can be considered as an alternative:

$$\hat{\boldsymbol{X}} = \underset{\boldsymbol{X} \succeq 0 \text{ and } tr(\boldsymbol{X}) = 1}{\arg\max}\ \langle \boldsymbol{M}, \boldsymbol{X} \rangle - \rho \|\boldsymbol{X}\|_{1,1}.$$

By letting $\boldsymbol{X} = \boldsymbol{x}\boldsymbol{x}^\top$, the equivalence of the above two objective functions can be easily justified. Since $supp(\boldsymbol{x}) = supp(diag(\boldsymbol{x}\boldsymbol{x}^\top))$, we estimate the support $J$ by $\hat{J} = supp(diag(\hat{\boldsymbol{X}}))$ in the semidefinite problem. This kind of relaxation has been studied by d'Aspremont et al. [2004] and Lei and Vu [2015], but their works were limited to complete data. Surprisingly, without any additional modifications on the relaxation problem such as using matrix factorization or matrix completion, we show that it is possible to exactly recover true support $J$ with the above semidefinite program itself when $\boldsymbol{M}$ is an incomplete observation. Our main contribution is to prove this claim theoretically and experimentally.

In Section 3, we provide theoretical justification (i.e., Theorem 1) that we can exactly recover the true support $J$ with high probability by obtaining a unique solution of the semidefinite problem, under proper conditions. The conditions involve matrix coherence parameters, the spectral gap between the largest and second-largest eigenvalues of the true matrix, the observation probability and the noise variance, which are discussed in detail in Corollaries 1 and 2. Specifically, we show that the sample complexity is related to the matrix coherence parameters as well as the matrix dimension $d$ and the support size $s$. We prove that the observation probability $p$ has the bound of $p = \omega\left(\frac{1}{d^{-1}+1}\right)$ in the worst scenario in terms of the matrix coherence, while it has a smaller lower bound $p = \omega\left(\frac{1}{(\log s)^{-1}+1}\right)$ in the best scenario. In Section 4, we provide experimental results on incomplete synthetic datasets and a gene expression dataset. The experiment on the synthetic datasets validate our theoretical results, and the experiment on the gene expression dataset gives us a consistent result with prior studies. We also show that our SDP algorithm outperforms several other sparse PCA approaches in the synthetic dataset.

## 2 Preliminaries

### 2.1 Notation

We first introduce the notations used throughout the paper. Matrices are bold capital, vectors are bold lowercase and scalars or entries are not bold. For any positive integer $n$, we denote $[n] := \{1, \ldots, n\}$. For any vector $\boldsymbol{a} \in \mathbb{R}^d$ and index set $J \subseteq [d]$, $\boldsymbol{a}_J$ denotes the $|J|$-dimensional vector consisting of

the entries of $\boldsymbol{a}$ in $J$. For any matrix $\boldsymbol{A} \in \mathbb{R}^{d_1 \times d_2}$ and index sets $J_1 \subseteq [d_1]$ and $J_2 \subseteq [d_2]$, $\boldsymbol{A}_{J_1,J_2}$ and $\boldsymbol{A}_{J_1,:}(\boldsymbol{A}_{:,J_2})$ denote the $|J_1| \times |J_2|$ sub-matrix of $\boldsymbol{A}$ consisting of rows in $J_1$ and columns in $J_2$, and the $|J_1| \times d_2$ ($d_1 \times |J_2|$) sub-matrix of $\boldsymbol{A}$ consisting of rows in $J_1$ (columns in $J_2$), respectively. $\|\boldsymbol{a}\|_1$, $\|\boldsymbol{a}\|_2$ and $\|\boldsymbol{a}\|_\infty$ represent the $l_1$ norm, $l_2$ norm and maximum norm of a vector $\boldsymbol{a}$, respectively. $\{\boldsymbol{e}_i \ : \ i \in [d]\}$ indicates the standard basis of $\mathbb{R}^d$.

A variety of norms on matrices will be used: we denote by $\|\boldsymbol{A}\|_2$ the spectral norm and by $\|\boldsymbol{A}\|_F$ the Frobenius norm of a matrix $\boldsymbol{A}$. We let $\|\boldsymbol{A}\|_{1,1} = \sum_{i \in [d_1], j \in [d_2]} |A_{i,j}|$, $\|\boldsymbol{A}\|_{\max} = \|\boldsymbol{A}\|_{\infty,\infty} = \max_{i \in [d_1], j \in [d_2]} |A_{i,j}|$, $\|\boldsymbol{A}\|_{2,\infty} = \max_{j \in [d_2]} \|\boldsymbol{A}_{:,j}\|_2$ and $\|\boldsymbol{A}\|_{1,\infty} = \max_{j \in [d_2]} \|\boldsymbol{A}_{:,j}\|_1$ represent the $l_{1,1}$ norm, the entrywise $l_\infty$ norm, the $l_{2,\infty}$ norm and the $l_{1,\infty}$ norm of a matrix $\boldsymbol{A}$, respectively. The trace of $\boldsymbol{A}$ is denoted $tr(\boldsymbol{A})$, and the matrix inner product of $\boldsymbol{A}$ and $\boldsymbol{B}$ is denoted $\langle \boldsymbol{A}, \boldsymbol{B} \rangle$. Also, $\sigma_i(\boldsymbol{A})$ and $\lambda_i(\boldsymbol{A})$ represent the $i$th largest singular value and the $i$th largest eigenvalue of $\boldsymbol{A}$, respectively.

The notation $C, C_1, \ldots, c, c_1, \ldots$ denote positive constants whose values may change from line to line. The notation $f(x) = o(g(x))$ or $f(x) \ll g(x)$ means $\lim_{x \to \infty} f(x)/g(x) = 0$; $f(x) = \omega(g(x))$ or $f(x) \gg g(x)$ means $\lim_{x \to \infty} f(x)/g(x) = \infty$; $f(x) = O(g(x))$ or $f(x) \lesssim g(x)$ means that there exists a constant $C$ such that $f(x) \leq Cg(x)$ asymptotically; $f(x) = \Omega(g(x))$ or $f(x) \gtrsim g(x)$ means that there exists a constant $C$ such that $f(x) \geq Cg(x)$ asymptotically; $f(x) = \Theta(g(x))$ or $f(x) \simeq g(x)$ means that there exists constants $C$ and $C'$ such that $Cg(x) \leq f(x) \leq C'g(x)$ asymptotically.

## 2.2 Model

We now introduce our model assumption. Suppose that an unknown matrix $\boldsymbol{M}^* \in \mathbb{R}^{d \times d}$ is symmetric. The spectral decomposition of $\boldsymbol{M}^*$ is given by

$$\boldsymbol{M}^* = \sum_{k \in [d]} \lambda_k(\boldsymbol{M}^*) \boldsymbol{u}_k \boldsymbol{u}_k^\top,$$

where $\lambda_1(\boldsymbol{M}^*) \geq \cdots \geq \lambda_d(\boldsymbol{M}^*)$ are its eigenvalues and $\boldsymbol{u}_1, \ldots, \boldsymbol{u}_d \in \mathbb{R}^d$ are the corresponding eigenvectors. We assume that $\lambda_1(\boldsymbol{M}^*) > \lambda_2(\boldsymbol{M}^*)$ and the leading eigenvector $\boldsymbol{u}_1$ of $\boldsymbol{M}^*$ is sparse, that is, for some set $J \in [d]$,

$$\begin{cases} u_{1,i} \neq 0 & \text{if } i \in J \\ u_{1,i} = 0 & \text{otherwise.} \end{cases}$$

With a notation $supp(\boldsymbol{a}) := \{i \in [d] : a_i \neq 0\}$ for any vector $\boldsymbol{a} \in \mathbb{R}^d$, we can write $J = supp(\boldsymbol{u}_1)$. Also, we denote the size of $J$ by $s$.

**Incomplete and noisy observation**  Suppose that we have only noisy observations of the entries of $\boldsymbol{M}^*$ over a sampling set $\Omega \subseteq [d] \times [d]$. Specifically, we observe a symmetric matrix $\boldsymbol{M} \in \mathbb{R}^{d \times d}$ such that

$$M_{i,j} = M_{j,i} = \delta_{i,j} \cdot (M_{i,j}^* + \epsilon_{i,j})$$

for $1 \leq i \leq j \leq d$, where $\delta_{i,j} = 1$ if $(i,j) \in \Omega$ and $\delta_{i,j} = 0$ otherwise, and $\epsilon_{i,j}$ is the noise at location $(i,j)$. In this paper, we consider the following assumptions on random sampling and random noise: for $1 \leq i \leq j \leq d$,

- Each $(i,j)$ is included in the sampling set $\Omega$ independently with probability $p$ (that is, $\delta_{i,j} \overset{i.i.d.}{\sim} Ber(p)$.)

- $\delta_{i,j}$'s and $\epsilon_{i,j}$'s are mutually independent.

- $\mathbb{E}[\epsilon_{i,j}] = 0$ and $\mathsf{Var}[\epsilon_{i,j}] = \sigma^2$.

- $|\epsilon_{i,j}| \leq B$ almost surely.

Finally, we define the coherence parameters of the sub-matrices $\boldsymbol{M}_{J,J}^*$, $\boldsymbol{M}_{J^c,J}^*$ and $\boldsymbol{M}_{J^c,J^c}^*$.

**Definition 1** (Coherence parameters). *We define the coherence parameters $\mu_0(\boldsymbol{M}^*_{J,J})$, $\mu_1(\boldsymbol{M}^*_{J,J})$, $\mu_2(\boldsymbol{M}^*_{J^c,J})$ and $\mu_3(\boldsymbol{M}^*_{J^c,J^c})$ as follows:*

$$\mu_0(\boldsymbol{M}^*_{J,J}) := \frac{\|\boldsymbol{M}^*_{J,J}\|_{\max}}{\lambda_1(\boldsymbol{M}^*_{J,J}) - \lambda_2(\boldsymbol{M}^*_{J,J})}, \quad \mu_1(\boldsymbol{M}^*_{J,J}) := \frac{\|\boldsymbol{M}^*_{J,J}\|_{\max}}{\|\boldsymbol{M}^*_{J,J}\|_{2,\infty}},$$

$$\mu_2(\boldsymbol{M}^*_{J^c,J}) := \frac{\|\boldsymbol{M}^*_{J^c,J}\|_{\max}}{\|\boldsymbol{M}^*_{J^c,J}\|_F}, \qquad \mu_3(\boldsymbol{M}^*_{J^c,J^c}) := \frac{\|\boldsymbol{M}^*_{J^c,J^c}\|_{\max}}{\max\left\{\|\boldsymbol{M}^*_{J^c,J^c}\|_2, \|\boldsymbol{M}^*_{J^c,J^c}\|_{2,\infty}\right\}}.$$

*We use $\mu_0$, $\mu_1$, $\mu_2$ and $\mu_3$ as shorthand for $\mu_0(\boldsymbol{M}^*_{J,J})$, $\mu_1(\boldsymbol{M}^*_{J,J})$, $\mu_2(\boldsymbol{M}^*_{J^c,J})$ and $\mu_3(\boldsymbol{M}^*_{J^c,J^c})$, respectively.*

Intuitively, when each coherence parameter is small, all the entries of the corresponding matrix have comparable magnitudes. Note that $\frac{1}{s} \leq \mu_0 \leq 1$, $\frac{1}{\sqrt{s}} \leq \mu_1 \leq 1$, $\frac{1}{\sqrt{s(d-s)}} \leq \mu_2 \leq 1$, $\frac{1}{d-s} \leq \mu_3 \leq 1$.

**Remark.** *We note that our setting is different from the case in which a data matrix $\boldsymbol{Y}$ is observed with missing entries and the covariance matrix $\boldsymbol{M} = \boldsymbol{Y}^\top \boldsymbol{Y}$ is analyzed for sparse PCA. Instead, we directly observe a general symmetric incomplete matrix $\boldsymbol{M}$, which is not necessarily positive semidefinite. Our model can be therefore applied to various data types other than covariance matrices (e.g., undirected random graphs with missing edges.) When a noisy and incomplete data matrix $\boldsymbol{Y}$ is given and its covariance matrix is considered, our current analysis cannot be directly applied and one may need some bias correction technique in the algorithm, since $\mathbb{E}[\boldsymbol{M}] = \mathbb{E}[\boldsymbol{Y}^\top \boldsymbol{Y}]$ is not the true matrix $\boldsymbol{Y}^{*\top} \boldsymbol{Y}^*$ where $\boldsymbol{Y}^* = \mathbb{E}[\boldsymbol{Y}]$. Furthermore, the independent missingness mechanism over $\boldsymbol{Y}$ does not imply independent missingness over $\boldsymbol{M}$. We leave this for future work.*

## 3 Main Results

As mentioned in the introduction, we consider the following semidefinite programming (SDP) in order to recover the true support $J$:

$$\hat{\boldsymbol{X}} = \underset{\boldsymbol{X} \succeq 0 \text{ and } tr(\boldsymbol{X})=1}{\arg\max} \langle \boldsymbol{M}, \boldsymbol{X} \rangle - \rho\|\boldsymbol{X}\|_{1,1}, \tag{1}$$

where we estimate $J$ by $\hat{J} = supp(diag(\hat{\boldsymbol{X}}))$. We recall that (1) is a convex relaxation of the following nonconvex problem:

$$\hat{\boldsymbol{x}} = \underset{\boldsymbol{x}^\top \boldsymbol{x}=1}{\arg\max} \ \boldsymbol{x}^\top \boldsymbol{M} \boldsymbol{x} - \rho\|\boldsymbol{x}\|_1^2. \tag{2}$$

In Theorem 1, we will show that under appropriate conditions, the solution of (1) attains $\hat{J} = J$ with high probability. Our main technical tool used in the proof is the primal-dual witness argument [Wainwright, 2009]. We start with deriving the sufficient conditions for the primal-dual solutions of (1) to be uniquely determined and satisfy $supp(diag(\hat{\boldsymbol{X}})) = J$. We then establish a proper candidate solution which meets the derived sufficient conditions, where we make use of the Karush-Kuhn-Tucker (KKT) conditions of (2) to set up a reasonable candidate. We finally develop the conditions under which the established candidate solution satisfies the sufficient conditions from the primal-dual witness argument of (1) with high probability. Detailed proof is given in Appendix B.

**Theorem 1.** *Under the model defined in Section 2.2, assume that the following conditions hold:*

$$2\sqrt{2} \cdot \frac{K_1 + \rho s}{p(\lambda_1(\boldsymbol{M}^*_{J,J}) - \lambda_2(\boldsymbol{M}^*_{J,J}))} \leq \min_{i \in J} |u_{1,i}|,$$

$$\rho > 2\sqrt{ps^c \cdot \left\{(1-p)\|\boldsymbol{M}^*_{J^c,J}\|_F^2 + (d-s)s\sigma^2\right\}} + p \cdot \|\boldsymbol{M}^*_{J^c,J}\|_{\max}$$

$$(K_2 + p \cdot \|\boldsymbol{M}^*_{J^c,J}\|_2)^2 \cdot (1 + \sqrt{s})^2 \leq \left\{p \cdot (\lambda_1(\boldsymbol{M}^*_{J,J}) - \lambda_2(\boldsymbol{M}^*_{J,J})) - 2 \cdot K_1 - 2\rho s\right\}$$

$$\times \left\{p \cdot (\lambda_1(\boldsymbol{M}^*_{J,J}) - \lambda_1(\boldsymbol{M}^*_{J^c,J^c})) - K_1 - K_3 - \rho d\right\},$$

*where $c > 0$, and $K_1$, $K_2$ and $K_3$ are defined as follows:*

$$K_1 := (c+1) \cdot R_1 \log(2s) + \sqrt{2(c+1)} \cdot R_2\sqrt{\log(2s)}$$

$$K_2 := (c+1) \cdot R_3 \log d + \sqrt{2(c+1)} \cdot R_4\sqrt{\log d}$$

$$K_3 := (c+1) \cdot R_5 \log(2(d-s)) + \sqrt{2(c+1)} \cdot R_6\sqrt{\log(2(d-s))}$$

*and*

$$R_1 := \max\{(1-p)\|\boldsymbol{M}^*_{J,J}\|_{\max} + B, \ p\|\boldsymbol{M}^*_{J,J}\|_{\max}\},$$

$$R_2 := \sqrt{p(1-p)}\|\boldsymbol{M}^*_{J,J}\|_{2,\infty} + \sqrt{ps\sigma^2},$$

$$R_3 := \max\{(1-p)\|\boldsymbol{M}^*_{J^c,J}\|_{\max} + B, \ p\|\boldsymbol{M}^*_{J^c,J}\|_{\max}\},$$

$$R_4 := \max\{\sqrt{p(1-p)}\|\boldsymbol{M}^*_{J^c,J}\|_{2,\infty} + \sqrt{p(d-s)\sigma^2}, \sqrt{p(1-p)}\|\boldsymbol{M}^*_{J,J^c}\|_{2,\infty} + \sqrt{ps\sigma^2}\},$$

$$R_5 := \max\{(1-p)\|\boldsymbol{M}^*_{J^c,J^c}\|_{\max} + B, \ p\|\boldsymbol{M}^*_{J^c,J^c}\|_{\max}\},$$

$$R_6 := \sqrt{p(1-p)}\|\boldsymbol{M}^*_{J^c,J^c}\|_{2,\infty} + \sqrt{p(d-s)\sigma^2}.$$

*Then the optimal solution $\hat{\boldsymbol{X}}$ to the problem (1) is unique and satisfies $supp(diag(\hat{\boldsymbol{X}})) = J$ with probability at least $1 - s^{-c} - d^{-c} - (2s)^{-c} - (2(d-s))^{-c}$.*

**Remark 1.** *Our proposed method actually guarantees more than support recovery. Under the sufficient conditions, the optimal solution $\hat{\boldsymbol{X}}$ can be expressed as $\hat{\boldsymbol{X}} = \begin{pmatrix} \hat{\boldsymbol{x}}\hat{\boldsymbol{x}}^\top & 0 \\ 0 & 0 \end{pmatrix}$, where $\hat{\boldsymbol{x}}$ satisfies $\left\|\boldsymbol{u}_{1,J} - \hat{\boldsymbol{x}}\right\|_2 \leq \min_{j\in J} |u_{1,j}|$ which is shown in our proof. This implies consistency of our solution in eigenvector estimation.*

**Remark 2.** *To deal with missingness, we utilized concentration inequalities such as matrix Bernstein inequality and Chebychev's inequality, which is one key technical difference from prior works on sparse PCA with complete data [Lei and Vu, 2015]. Beyond this, under the primal-dual witness framework, while Lei and Vu [2015] employed implicit constraints in the optimization problem, we made all the constraints explicit and directly derived the sufficient conditions to avoid strong conditions. We note that simply applying concentration inequalities to the approach of Lei and Vu [2015] results in stricter requirements on $\boldsymbol{M}^*_{J^c,J^c}$ and $p$.*

We now consider the following two particular scenarios:

(s1) $B = \sigma_2 = 0$, that is, the observation $\boldsymbol{M}$ is noiseless (but still incomplete).

(s2) The rank of $\boldsymbol{M}^*$ is 1.

to better interpret the conditions of $\boldsymbol{M}^*$ and $p$ listed in Theorem 1 and understand under what circumstance these conditions hold. For both cases, we set $p \geq 0.5$ for simplicity. Under the first setting, we can re-express the conditions on $\boldsymbol{M}^*$ for exact sparse recovery of $J$ in a more interpretable way (specifically, in terms of coherence parameters and spectral gap) as well as the conditions on $p$. In the second setting, we aim to investigate that the maximum level of noise that is allowed by Theorem 1. Corollaries 1 and 2 include the results of the two settings (s1) and (s2), respectively.

**Corollary 1.** *Assume that $B = \sigma_2 = 0$, $p \geq 0.5$ and $\min_{i\in J} |u_{1,i}| = \Omega(\frac{1}{\sqrt{s}})$. Denote $\lambda_1(\boldsymbol{M}^*_{J,J}) - \lambda_2(\boldsymbol{M}^*_{J,J})$ by $\bar{\lambda}(\boldsymbol{M}^*_{J,J})$. If the following conditions hold:*

$$\mu_0 = o\left(\frac{1}{\sqrt{s}\log s}\right), \tag{3}$$

$$\|\boldsymbol{M}^*_{J^c,J}\|_{\max} = o\left(\frac{\bar{\lambda}(\boldsymbol{M}^*_{J,J})}{s} \cdot \min\left\{\mu_2, \frac{1}{s}, \frac{\sqrt{s}}{\log d}\right\}\right), \tag{4}$$

$$\|\boldsymbol{M}^*_{J^c,J^c}\|_{\max} = o\left(\bar{\lambda}(\boldsymbol{M}^*_{J,J}) \cdot \min\left\{\mu_3, \frac{1}{\log(d-s)}\right\}\right), \tag{5}$$

$$\sqrt{\frac{1-p}{p}} = o\bigg(\min\Big\{\mu_1\sqrt{\log s},$$
$$\frac{\bar{\lambda}(\boldsymbol{M}^*_{J,J})\mu_2}{\|\boldsymbol{M}^*_{J^c,J}\|_{\max}} \cdot \min\left\{\frac{1}{s^2\sqrt{s}}, \frac{1}{s\sqrt{s(d-s)}}\right\}, \tag{6}$$
$$\frac{\bar{\lambda}(\boldsymbol{M}^*_{J,J})\mu_3}{\|\boldsymbol{M}^*_{J^c,J^c}\|_{\max}} \cdot \frac{1}{\sqrt{\log(d-s)}}\Big\}\bigg),^1$$

$$\rho = \Theta\left(\frac{p\bar{\lambda}(\boldsymbol{M}^*_{J,J})}{s^2}\right), \tag{7}$$

*then the conditions in Theorem 1 hold asymptotically, that is, when $s$ and $d$ are sufficiently large, the optimal solution $\hat{X}$ to the problem (1) is unique and satisfies $supp(diag(\hat{X})) = J$ with probability at least $1 - s^{-1} - d^{-1} - (2s)^{-1} - (2(d-s))^{-1}$.*

**Conditions on true matrix $M^*$**    From the conditions in Corollary 1, we can find desirable properties on the matrix $M^*$ as follows:

- *Incoherence of $M^*_{J,J}$, and coherence of $M^*_{J^c,J}$ and $M^*_{J^c,J^c}$*: From the coherence parameter in (3) and those in (4), (5) and (6), we see that the sub-matrix $M^*_{J,J}$ and the sub-matrices $M^*_{J^c,J}$ and $M^*_{J^c,J^c}$ are expected to be incoherent and coherent, respectively. This is different from other problems involving incomplete matrices, such as matrix completion [Candès and Recht, 2009] and standard PCA on incomplete data [Cai et al., 2021], where the entire matrix, not a sub-matrix, is required to be incoherent.

  We can easily check the need of incoherence of $M^*_{J^c,J}$ with an example that the sub-matrix has only one entry with a large magnitude while the other entries have relatively small values. Even if the true leading eigenvector of the sub-matrix is not sparse, the sparse PCA algorithm may produce a solution $\hat{J}$ which has a smaller size than that of the true support $J$. However, for $M^*_{J^c,J}$ and $M^*_{J^c,J^c}$, coherence is preferable: intuitively speaking, when $M^*_{J^c,J}$ and $M^*_{J^c,J^c}$ are the most coherent, that is, only one entry is nonzero in each sub-matrix, and all other entries are zero, missing the entries in $M^*_{J^c,J}$ and $M^*_{J^c,J^c}$ does not change the leading eigenvector of $M^*$. On the other hand, when $M^*_{J^c,J}$ and $M^*_{J^c,J^c}$ are incoherent, that is, all the entries have comparable magnitudes, missing only a few entries changes the leading eigenvector and its sparsitency, so that sparse PCA is likely to fail to recover $J$. A simple illustration can be found in the Appendix A.

- *Large spectral gap $\bar{\lambda}(M^*_{J,J})$ $(= \lambda_1(M^*_{J,J}) - \lambda_2(M^*_{J,J}))$*: This can be found in (4), (5) and (6). A sufficiently large spectral gap requirement has been also discussed in the work on sparse PCA on the complete matrix [Lei and Vu, 2015]. It ensures the uniqueness and identifiability of the orthogonal projection matrix with respect to the principal subspace. If the spectral gap of eigenvalues is nearly zero, then the top two eigenvectors are indistinguishable given the observational noise, leading to failure to recover the sparsity of the leading eigenvector.
  We also note that $\lambda_1(M^*_{J,J}) - \lambda_2(M^*_{J,J}) \geq \lambda_1(M^*) - \lambda_2(M^*)$ since $\lambda_1(M^*_{J,J}) = \lambda_1(M^*)$ and $\lambda_2(M^*_{J,J}) \leq \lambda_2(M^*)$. Hence, a large $\lambda_1(M^*) - \lambda_2(M^*)$ implies a large $\bar{\lambda}(M^*_{J,J})$.

- *Small magnitudes of $M^*_{J^c,J}$ and $M^*_{J^c,J^c}$*: This can also be found in (4), (5) and (6). This condition is also natural: if the magnitudes are relatively small, missing the entries will not make a big impact to the result.

**Conditions on $p$ (ratio of missing data)**    For simplicity, suppose that $\bar{\lambda}(M^*_{J,J}) = O(s)$ and $s = O(\log d)$. We consider two extreme cases where the coherence parameters are maximized and minimized. We discuss the bound of the sample complexity in each case.

- *The best scenario where the bound of the sample complexity is the lowest*: Suppose that $\mu_1 = o(\frac{1}{\log s})$ and $\mu_2 = \mu_3 = 1$ (note that when $\mu_0 = o(\frac{1}{\sqrt{s}\log s})$, $\mu_1$ is upper bounded by $o(\frac{1}{\log s})$.) Then the condition (6) can be written as:

$$p = \frac{1}{1 + \epsilon_1}, \quad \epsilon_1 = o\left( \min\left\{ \frac{1}{\log s}, \ \frac{1}{sd\|M^*_{J^c,J}\|^2_{\max}} \right\} \right).$$

  Furthermore, if $\|M^*_{J^c,J}\|_{\max} = O(\sqrt{\frac{\log s}{sd}})$, then $p = \omega\left( \frac{1}{(\log s)^{-1} + 1} \right)$.

---

[1]This can be expressed equivalently as $p = \frac{1}{1+\epsilon^2}$ where $\epsilon = o\left( \min\left\{ \mu_1\sqrt{\log s}, \ \frac{\bar{\lambda}(M^*_{J,J})\mu_2}{\|M^*_{J^c,J}\|_{\max}} \cdot \right. \right.$
$\min\left\{ \frac{1}{s^2\sqrt{s}}, \ \frac{1}{s\sqrt{s(d-s)}} \right\}, \ \frac{\bar{\lambda}(M^*_{J,J})\mu_3}{\|M^*_{J^c,J^c}\|_{\max}} \cdot \frac{1}{\sqrt{\log(d-s)}} \right\}$.

- *The worst scenario where the bound of the sample complexity is the highest*: Suppose that $\mu_1 = \frac{1}{\sqrt{s}}$, $\mu_2 = \frac{1}{\sqrt{s(d-s)}}$ and $\mu_3 = \frac{1}{d-s}$. In this case, the condition (6) can be written as:

$$p = \frac{1}{1+\epsilon_2}, \quad \epsilon_2 = o\left(\min\left\{\frac{\log s}{s}, \frac{1}{(sd\|\boldsymbol{M}^*_{J^c,J}\|_{\max})^2}, \frac{1}{(d\|\boldsymbol{M}^*_{J^c,J^c}\|_{\max})^2}\right\}\right).$$

Suppose that $\|\boldsymbol{M}^*_{J^c,J}\|_{\max} = \Theta(\frac{1}{s\sqrt{d}})$ and $\|\boldsymbol{M}^*_{J^c,J^c}\|_{\max} = \Theta(\frac{1}{\sqrt{d}})$ for instance, then $p = \omega\left(\frac{1}{d^{-1}+1}\right)$.

Next, we consider the second setting (s2) where the rank of $\boldsymbol{M}^*$ is assumed to be 1, that is, $\boldsymbol{M}^* = \lambda_1(\boldsymbol{M}^*)\boldsymbol{u}_1\boldsymbol{u}_1^\top$ (without loss of generality, we assume $\lambda_1(\boldsymbol{M}^*) > 0$). Trivially, $\boldsymbol{M}^*_{J^c,J} = \boldsymbol{M}^*_{J,J^c} = \boldsymbol{M}^*_{J^c,J^c} = 0$ and Theorem 1 can be greatly simplified. Here, we focus on analyzing how much noise (parameters $B$ and $\sigma^2$) is allowed.

**Corollary 2.** *Assume that $p \geq 0.5$ and the rank of $\boldsymbol{M}^*$ is 1, that is, $\boldsymbol{M}^* = \lambda_1(\boldsymbol{M}^*)\boldsymbol{u}_1\boldsymbol{u}_1^\top$. Let $\lambda_1(\boldsymbol{M}^*) > 0$. Suppose that $s$ and $d$ satisfy $\frac{1}{\sqrt{s}} \leq \frac{12+\frac{d-s}{s}+8\sqrt{2}a_2-\sqrt{(4-\frac{d-s}{s}-8\sqrt{2}a_2)^2+512a_1^2(1+\sqrt{s})^2}}{4\sqrt{2}+\sqrt{2}\cdot\frac{d-s}{s}+16a_2-16\sqrt{2}a_1^2(1+\sqrt{s})^2}$ where $a_1 = (2-\frac{1}{p})\cdot\frac{\log d}{8\sqrt{2}\log(2s)} + \frac{\sqrt{\max\{d-s,s\}}\cdot\sqrt{\log d}}{16s^2\sqrt{d-s}}$ and $a_2 = (2-\frac{1}{p})\cdot\frac{\log(2(d-s))}{8\sqrt{2}\log(2s)} + \frac{\sqrt{\log(2(d-s))}}{16s^2}$. If the following conditions hold:*

$$\frac{\max_{i,j\in J}|u_{1,i}u_{1,j}|}{\min_{i\in J}|u_{1,i}|} \leq \frac{1}{16\sqrt{2}\log(2s)},$$

$$\frac{\max_{i\in J}|u_{1,i}|}{\min_{i\in J}|u_{1,i}|} \leq \frac{1}{16\sqrt{2}\sqrt{\log(2s)}}\cdot\sqrt{\frac{p}{1-p}},$$

$$B \leq (2p-1)\lambda_1(\boldsymbol{M}^*)\cdot\max_{i,j\in J}|u_{1,i}u_{1,j}|,$$

$$2\sqrt{2}\cdot\sqrt{p\sigma^2 s^2(d-s)} < \rho \leq \frac{1}{8\sqrt{2s}}\cdot p\lambda_1(\boldsymbol{M}^*)\cdot\min_{i\in J}|u_{1,i}|,$$

*then the optimal solution $\hat{\boldsymbol{X}}$ to the problem (1) is unique and satisfies $supp(diag(\hat{\boldsymbol{X}})) = J$ with probability at least $1 - s^{-1} - d^{-1} - (2s)^{-1} - (2(d-s))^{-1}$.*

**Conditions on noise parameters $B$ and $\sigma^2$** For simplicity, let $\lambda_1(\boldsymbol{M}^*) = O(s)$ and $\forall |u_{1,i}| = \Theta(\frac{1}{\sqrt{s}})$. Then the above conditions in Corollary 2 imply that

$$B \lesssim p \quad \text{and} \quad \sigma^2 \lesssim \frac{p}{s^3(d-s)}.$$

The condition for $B$ is relatively moderate while $\sigma^2$ needs to be extremely small to satisfy the condition in Corollary 2. We comment this is only a sufficient condition, and the experimental results show that (1) can succeed even with $\sigma^2$ larger than the aforementioned bound.

## 4 Numerical Results

We perform the SDP algorithm of (1) on synthetic and real data to validate our theoretic results and show how well the true support of the sparse principal component is exactly recovered. We also compare our SDP algorithm with other sparse PCA methods and show that our method performs better. Our experiments were executed on `MATLAB` and standard `CVX` code was used, although very efficient scalable SDP solvers exist in practice [Yurtsever et al., 2021].

### 4.1 Synthetic Data

We perform two lines of experiments:

1. With the spectral gap $\lambda_1(\boldsymbol{M}^*) - \lambda_2(\boldsymbol{M}^*)$ and the noise parameters $B$ and $\sigma^2$ fixed, we compare the results for different $s$ and $d$.

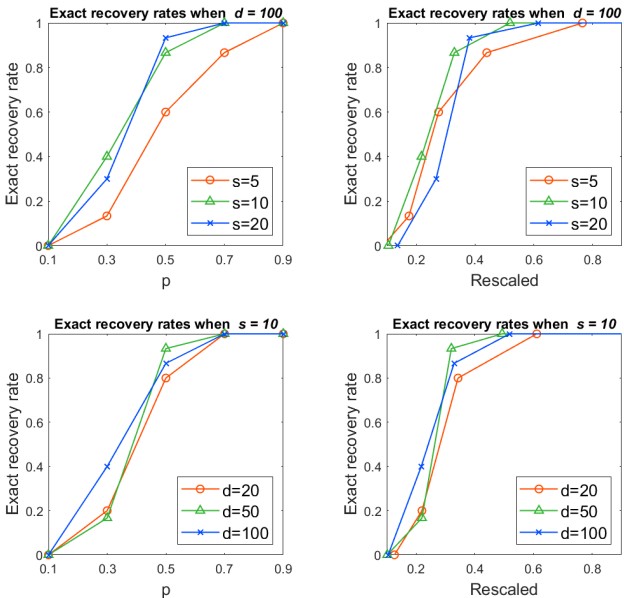

Figure 1: Results of experiment 1 on synthetic data.

2. With $s$ and $d$ fixed, we compare the results for different spectral gaps and noise parameters.

In each experiment, we generate the true matrix $\boldsymbol{M}^*$ as follows: the leading eigenvector $\boldsymbol{u}_1$ is set to have $s$ number of non-zero entries. $\lambda_2(\boldsymbol{M}^*), \dots, \lambda_d(\boldsymbol{M}^*)$ are randomly selected from a normal distribution with mean 0 and standard deviation 1, and $\lambda_1(\boldsymbol{M}^*)$ is set to $\lambda_2(\boldsymbol{M}^*)$ plus the spectral gap. The orthogonal eigenvectors are randomly selected, while the non-zero entries of the leading eigenvector $\boldsymbol{u}_1$ are made to have a value of at least $\frac{1}{2\sqrt{s}}$.

When generating the observation $\boldsymbol{M}$, we first add to $\boldsymbol{M}^*$ the entry-wise noise which is randomly selected from a truncated normal distribution with support $[-B, B]$. The normal distribution to be truncated is set to have mean 0 and standard deviation $\sigma_{normal}$. After adding the entry-wise noise, we generate an incomplete matrix $\boldsymbol{M}$ by selecting the observed entries uniformly at random with probability $p \in \{0.1, 0.3, 0.5, 0.7, 0.9\}$.

In each setting, we run the algorithm (1) and verify if the solution exactly recovers the true support. We repeat each experiment 30 times with different random seeds, and calculate the rate of exact recovery in each setting.

**Experiment 1** In this experiment, we fix the spectral gap $\lambda_1(\boldsymbol{M}^*) - \lambda_2(\boldsymbol{M}^*)$ as 20 and the noise parameters $B$ and $\sigma^2$ as 5 and 0.01. We use the tuning parameter $\rho = 0.1$. We try three different matrix dimensions $d \in \{20, 50, 100\}$ and three different support sizes $s \in \{5, 10, 20\}$.

To check whether the bound of the sample complexity obtained in Corollary 1 is tight, we calculate the coherence parameters and the maximum magnitudes of the sub-matrices at each setting, and calculate the following rescaled parameter:

$$\sqrt{\frac{p}{1-p}} \cdot \min \left\{ \mu_1 \sqrt{\log s}, \frac{\bar{\lambda}(\boldsymbol{M}^*_{J,J})\mu_2}{\|\boldsymbol{M}^*_{J^c,J}\|_{\max}} \cdot \min \left\{ \frac{1}{s^2\sqrt{s}}, \frac{1}{s\sqrt{s(d-s)}} \right\}, \frac{\bar{\lambda}(\boldsymbol{M}^*_{J,J})\mu_3}{\|\boldsymbol{M}^*_{J^c,J^c}\|_{\max}} \cdot \frac{1}{\sqrt{\log(d-s)}} \right\},$$

which is derived from (6). If the exact recovery rate versus this rescaled parameter is the same across different settings, then we empirically justify that the bound of the sample complexity we derive is "tight"[2] in the sense that the exact recovery rate is solely determined by this rescaled parameter.

Figure 1 shows the experimental results. The two plots above are the experimental results for different values of $s$ when $d = 100$, and the two plots below are for different values of $d$ when $s = 10$.

---

[2] We note that "tight" does not mean an agreement between necessary and sufficient conditions here.

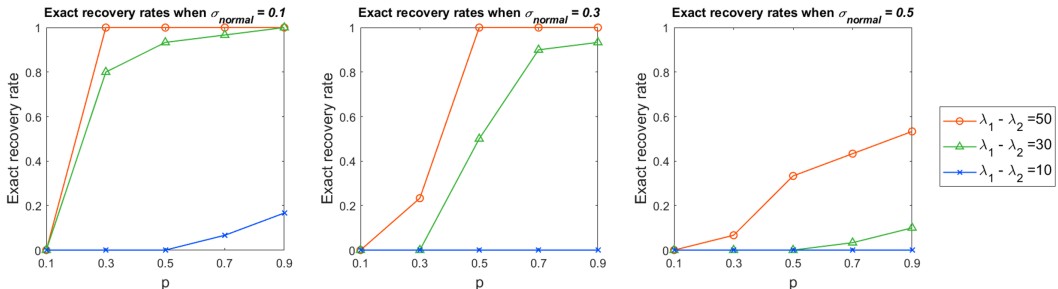

Figure 2: Results of experiment 2 on synthetic data.

The x-axis of the left graphs represents $p$, and the x-axis of the right graphs indicates the rescaled parameter.

We can see from the two graphs on the right that the exact recovery rate versus the rescaled parameter is the same in different settings of $d$ and $s$. This means that our bound of the sample complexity is tight.

Another observation we can make is that the exact recovery rate is not necessarily increasing or decreasing as $s$ or $d$ increases or decreases. This is probably because coherences and maximum magnitudes of sub-matrices are involved in the sample complexity as well.

**Experiment 2**   Here, we fix the matrix dimension $d$ as 100 and the support size $s$ as 50. We set $B = 5$. We try three different spectral gaps $\lambda_1(\boldsymbol{M}^*) - \lambda_2(\boldsymbol{M}^*) \in \{10, 30, 50\}$ and three different standard deviations of the normal distribution, $\sigma_{normal} \in \{0.1, 0.3, 0.5\}$. We try two different tuning parameters $\rho \in \{0.1, 0.01\}$ and report the best result.

Figure 2 demonstrates the experimental results. The three plots show the results when $\sigma_{normal}$ is 0.1, 0.3 and 0.5, respectively. The red, green and blue lines indicate the cases where the spectral gap $\lambda_1(\boldsymbol{M}^*) - \lambda_2(\boldsymbol{M}^*)$ is 50, 30 and 10, respectively. From the plots, we can observe that the exact recovery rate increases as $\sigma^2$ is small and $\lambda_1(\boldsymbol{M}^*) - \lambda_2(\boldsymbol{M}^*)$ is large, which is consistent with the conditions we have checked in Corollaries 1 and 2.

### 4.2   Gene Expression Data

We analyze a gene expression dataset (GSE21385) from the Gene Expression Omnibus website (`https://www.ncbi.nlm.nih.gov/geo/`.) The dataset examines rheumatoid arthritis synovial fibroblasts, which together with synovial macrophages, are the two leading cell types that invade and degrade cartilage and bone.

The original data set contains 56 subjects and 112 genes. We compute its incomplete covariance matrix, where 87% of the matrix entries are observed since some subject/gene pairs are unobserved. With this incomplete covariance matrix, we solve the semidefinite program in (1) for sparse PCA with $\rho = 2$.

By solving (1), we find that the support of the solution contains 3 genes: beta-1 catenin (CTNNB), hypoxanthine-guanine phosphoribosyltransferase 1 (HPRT1) and semaphorin III/F (SEMA3F). Our result is consistent with prior studies on rheumatoid arthritis since CTNNB has been found to be upregulated [Iwamoto et al., 2018], SEMA3F has been found to be downregulated [Tang et al., 2018], and HPRT1 is known to be a housekeeping gene [Mesko et al., 2013]. Additional illustration of this application can be found in Appendix G.

### 4.3   Comparison with Other Methods

We compare our SDP algorithm with three different methods. First, we consider two sparse PCA algorithms where missing cells are treated as zero: the diagonal thresholding sparse PCA (DTSPCA) by Johnstone and Lu [2009] and the iterative thresholding sparse PCA (ITSPCA) by Ma [2013]. Second, we consider the combination of imputation and our SDP method: we first estimate the

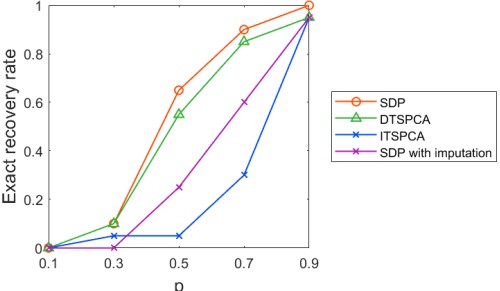

Figure 3: Comparison with other methods.

missing elements of $\boldsymbol{M}$ by using the matrix completion method based on the following Frobenius norm miminization with nuclear norm penalty $\|\boldsymbol{Y}\|_*$:

$$\tilde{\boldsymbol{M}} = \underset{\boldsymbol{Y}:symmetric}{\arg\min} \|P_\Omega(\boldsymbol{Y}) - \boldsymbol{M}\|_F^2 + \tau\|\boldsymbol{Y}\|_*$$

and then implemented the SDP method with the completed matrix $\tilde{\boldsymbol{M}}$. Details of the simulation settings and algorithm implementations are provided in Appendix F.

Figure 3 demonstrates that the exact recovery rate of our SDP method is higher than those of the three other methods. The comparison shows that the imputation-based strategy fails to work under the small-$p$ regime, and we conjecture the following rationale: it is known that the matrix completion algorithm can be successful under the low-rank assumption. However, we do not assume that the true matrix is low-rank, but just that its spectral gap $\lambda_1(\boldsymbol{M}^*) - \lambda_2(\boldsymbol{M}^*)$ is large enough, which probably leads to the failure of matrix completion. Over the 20 simulation replications, we found that the average of $\max_{i,j}|\tilde{M}_{ij} - M_{ij}^*|$ was 1.0260 while the averages of $\max_{i,j}|M_{ij}^*|$ and $\min_{i,j}|M_{ij}^*|$ are 2.3200 and 0.0002, respectively. This shows that the matrix completion has not been done very precisely. Given such an unsuccessful matrix completion, the imputed cells introduce more noise into the inference and the result of sparse PCA can be even worse than that of simply using zero for the missing entries.

## 5   Concluding Remarks

We have presented the sufficient conditions to exactly recover the true support of the sparse leading eigenvector by solving a simple semidefinite programming on an incomplete and noisy observation. We have shown that the conditions involve matrix coherence, spectral gap, matrix magnitudes, sample complexity and variance of noise, and provided empirical evidence to justify our theoretical results. To the best of our knowledge, we provide the first theoretical guarantee for exact support recovery with sparse PCA on incomplete data. While we currently focus on a uniformly missing at random setup, an interesting open question is whether it is possible to provide guarantees for a deterministic pattern of missing entries.

## Acknowledgments

This material is based upon work supported by the National Science Foundation under Grant No. 2134209-DMS.

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
