# OpenReview forum: "Support Recovery in Sparse PCA with Incomplete Data"
_NeurIPS.cc/2022/Conference — NeurIPS 2022 Accept_

### Official Review · Reviewer_oQV2 · 2022-06-27

**Rating:** 7
**Confidence:** 4
**Soundness:** 4 excellent
**Presentation:** 4 excellent
**Contribution:** 4 excellent

**Summary:**

This paper studies sparse PCA with incomplete and noisy data, and proposes a SDP relaxation algorithm. This relaxation formulation was previously applied to sparse PCA with complete data, and this paper shows that the same exact formulation can be applied to incomplete observations. Sufficient conditions for exact support recovery are proposed, and experiments on both synthetic and real-world data.

**Questions:**

- This is likely out of scope for this paper, but I am curious about how the proposed SDP method compares with other possible approaches to solving sparse PCA with incomplete data, such as various imputation methods. Such an experiment that evaluates the performance of the proposed method will further enhance the results of this paper.
- It may be helpful to expand on the gene expression data experiment, by showing some relevant visualizations of the data and/or results, for example in the Appendix.

**Limitations:**

Yes.

**Strengths And Weaknesses:**

- Overall, this is a well-written paper. It discusses the problem of sparse PCA with incomplete data, and presents a solid and novel theoretical approach to it via SDP relaxation.
- The experiment on gene expression data is another highlight, though it feels somewhat insufficiently illustrated.
- I don't see any major weaknesses of this paper.

---

> ### Author Response · Authors · 2022-08-02
> **Response to Reviewer oQV2**
>
> We thank the reviewer for the positive feedback and helpful suggestions.
>
> **How the proposed SDP method compares with other possible approaches to solving sparse PCA with incomplete data, such as various imputation methods:**
>
> We tried to first estimate the missing elements of $\pmb{M}$ by using the matrix completion method based on the following Frobenius norm miminization with nuclear norm penalty $|| \pmb{Y}||_*$:
>
> $\tilde{\pmb{M}} = \underset{\pmb{Y}: symmetric}{\arg\min} \Big( || P_{\Omega}(\pmb{Y}) - \pmb{M} ||_F^2$
>
> $~~~~+ \tau|| \pmb{Y} ||_* \Big)$
>
> and then implement the SDP method with the completed matrix $\tilde{\pmb{M}}$.
> Surprisingly, we could find that the method with imputation results in worse exact recovery rate than that of our method simply putting $0$ in the missing entries.
> We added Figure 4 illustrating this result in Section F of the appendix.
>
> We make the following conjecture for the reason why this happens:
> it is known that the matrix completion problem can be successful under low-rank assumption.
> However, we do not assume that the true matrix is low-rank, but just assume that its spectral gap $\lambda_1(\pmb{M}^*)-\lambda_2(\pmb{M}^*)$ is large enough.
> The randomly-generated true matrices in the experiment are not low-rank as well, which probably leads to the failure of matrix completion.
> We checked that the average of $\max_{i,j}|\tilde{M}_{ij}$
>
> $- M^*_{ij}|$
> was $1.0260$ while the averages of $\max_{i,j}|M^*_{ij}|$ and $\min_{i,j}|M^*_{ij}|$ are $2.3200$ and $0.0002$, respectively, in our simulation.
> This shows that the matrix completion has not been done very precisely.
>
>
> What we can check from the above experimental result is that
> if we do not have a very accurate imputation of the true matrix,
> the imputed cells introduce more noise into the inference and
> the result of sparse PCA can be even worse than that of simply using $0$ for the missing entries.
>
>
>
> **It may be helpful to expand on the gene expression data experiment, by showing some relevant visualizations of the data and/or results, for example in the Appendix.:**
>
> We added the heatmap of the incomplete covariance matrix of the gene expression data used in 4.2 to the appendix (see Figure 5 in Section G.)
> A sparse eigenvector with a large eigenvalue should lead to a block in the covariance matrix with large values.
> We can check this from the heatmap: the red box indicates the submatrix whose rows and columns correspond to the 3 genes (CTNNB,
> HPRT1 and SEMA3F) selected from the sparse PCA method.
> We observe that two of the three off-diagonals of this submatrix have significantly larger values than the other elements of the covariance matrix.

---

### Official Review · Reviewer_owug · 2022-07-07

**Rating:** 5
**Confidence:** 2
**Soundness:** 3 good
**Presentation:** 2 fair
**Contribution:** 2 fair

**Summary:**

The paper considers the sparse principal component analysis problem for incomplete and noisy data. In particular, it considers its semidefinite programming reformulated problem and develops a new algorithm with theoretical discussions on the uniqueness of the optimal solution with large probability. Numerical justification including the experiments on synthetic and realistic gene data sets have demonstrated the effectiveness. The work looks interesting with insightful theories.

**Questions:**

1. In Section 2.2, do the eigenvectors $\mathbf{u}_i$'s need to be normalized?
2. In the SDP reformulation in p.2, does it guarantee that the solution is rank-one and even $X=\mathbf{x}\mathbf{x}^T$?
3. In Theorem 1, what is $p$? How is it defined? Also, the existence of the optimal solution is not discussed.

**Limitations:**

The sparse principal component problem assumes that the optimal solution has sparse leading eigenvector which may not be the case in real applications and thus limit its applicability.

**Strengths And Weaknesses:**

Strengths: In the main result, several conditions have been proposed to guarantee the success of the SDP algorithm. There are some theoretical contributions. The application of gene expression is interesting and useful.

Weaknesses: The proposed algorithm is based on the SDP reformulation of the original sparse PCA problem, and the algorithmic development may be limited. In Section 4.1, there are no other related methods being compared.

---

> ### Author Response · Authors · 2022-08-02
> **Response to Reviewer owug (2)**
>
> **In Section 2.2, do the eigenvectors $\pmb{u_i}$'s need to be normalized?:**
>
> Yes, all the eigenvectors are unit vectors here.
> The signs of the eigenvectors $\pmb{u_i}$'s can be globally flipped, but that is not a big issue in our problem since we only focus on the support recovery.
> We also note that we assume $\lambda_1(\pmb{M}^*)>\lambda_2(\pmb{M}^*)$, and with this assumption, the leading eigenvector $\pmb{u_1}$ is uniquely determined, without regard to the sign.
>
> **In the SDP reformulation in p.2, does it guarantee that the solution is rank-one and even $\pmb{X} = \pmb{x}\pmb{x}^\top$?:**
>
> Yes. Under the sufficient conditions, the optimal solution $\hat{\pmb{X}}$ can be expressed as
> $\hat{\pmb{X}} = [\hat{\pmb{x}} \hat{\pmb{x}}^\top,~ \pmb{0};~ \pmb{0},~ \pmb{0}]$ and this is unique.
> The form of $\hat{\pmb{x}}$ is specified in Proposition 2 of Section B in the appendix.
> (Here $J$ is assumed to be $[s]$ without loss of generality, and for $J\neq[s]$, one needs to properly interleave matrix entries of $[\hat{\pmb{x}} \hat{\pmb{x}}^\top,~ \pmb{0};~ \pmb{0},~ \pmb{0}]$.)
>
>
> **In Theorem 1, what is $p$? How is it defined? Also, the existence of the optimal solution is not discussed.:**
>
> $p$ is the probability of each element of the true matrix $\pmb{M}^*$ being observed. Please check Section 2.2.
>
>
> Recall that our SDP problem is convex, and KKT conditions are sufficient and necessary for optimality.
> For convex optimization problem, the existence of the optimal solution is guaranteed if there are primal-dual solutions satisfying its KKT conditions.
> In the proof of Theorem 1, we specified a set of primal-dual solutions which satisfies the KKT conditions of our SDP problem.
> Hence, the existence of the optimal solution is guaranteed.
> The primal-dual solutions are introduced in Proposition 2 of Section B in the appendix.
>
>
>
> **The sparse principal component problem assumes that the optimal solution has sparse leading eigenvector which may not be the case in real applications and thus limit its applicability.:**
>
> We respectfully disagree with this statement, since the sparse PCA problem has already been widely studied in the setting without missingness (d’Aspremont et al. (2004), Amini and Wainwright (2009), Journee et al. (2010), Berthet and Rigollet (2013) and Yuan and Zhang (2013)),
> and leads to great successes in applications such as genome wide association studies and image compression.
>
>
> d'Aspremont, A., Ghaoui, L., Jordan, M., \& Lanckriet, G. (2004). A direct formulation for sparse PCA using semidefinite programming. Advances in neural information processing systems, 17.
>
> Amini, A. A., \& Wainwright, M. J. (2008, July). High-dimensional analysis of semidefinite relaxations for sparse principal components. In 2008 IEEE international symposium on information theory (pp. 2454-2458). IEEE.
>
> Journée, M., Nesterov, Y., Richtárik, P., \& Sepulchre, R. (2010). Generalized power method for sparse principal component analysis. Journal of Machine Learning Research, 11(2).
>
> Berthet, Q., \& Rigollet, P. (2013). Optimal detection of sparse principal components in high dimension. The Annals of Statistics, 41(4), 1780-1815.
>
> Yuan, X. T., \& Zhang, T. (2013). Truncated Power Method for Sparse Eigenvalue Problems. Journal of Machine Learning Research, 14(4).

---

> ### Author Response · Authors · 2022-08-02
> **Response to Reviewer owug (1)**
>
> We thank the reviewer for the assessment and helpful comments.
>
> **The proposed algorithm is based on the SDP reformulation of the original sparse PCA problem, and the algorithmic development may be limited.:**
>
> The advantage of the SDP algorithm is that it is simple and easy to implement.
> Also, even with its simplicity, it theoretically guarantees the exact support recovery and shows good experimental results.
> It shows good performance compared to other methods for sparse PCA as well. Please check the next paragraphs for this.
>
>
> **There are no other related methods being compared.:**
>
> We first compared our SDP method to two different algorithms: the diagonal thresholding sparse PCA (DTSPCA) by Johnstone and Lu (2009) and the iterative thresholding sparse PCA (ITSPCA) by Ma (2013) where missing cells are treated as 0.
> For the simulation, we set $d=50$, $s=10$, $\lambda_1(\pmb{M}^*)-\lambda_2(\pmb{M}^*) = 10$, $B=5$ and $\sigma_{normal} = 0.1$, and $20$ runs were computed.
>
> We added Figure 3 demonstrating the result in Section F of the appendix.
> From the figure, we can check that the exact recovery rate of SDP method is higher than those of DTSPCA and ITSPCA methods.
>
> Moreover, we tried the combination of imputation and our SDP method as well.
> We first estimated the missing elements of $\pmb{M}$ by using the matrix completion method based on the following Frobenius norm miminization with nuclear norm penalty $|| \pmb{Y}||_*$:
>
> $\tilde{\pmb{M}} = \underset{\pmb{Y}: symmetric}{\arg\min} \bigg( || P_{\Omega}(\pmb{Y}) - \pmb{M}||_F^2$
>
> $+ \tau|| \pmb{Y}||_* \bigg)$
>
> and then implemented the SDP method with the completed matrix $\tilde{\pmb{M}}$.
> Surprisingly, we could find that the method with imputation results in worse exact recovery rate than that of our method simply putting $0$ in the missing entries.
> We added Figure 4 illustrating this result in Section F of the appendix.
>
> We make the following conjecture for the reason why this happens:
> it is known that the matrix completion problem can be successful under low-rank assumption.
> However, we do not assume that the true matrix is low-rank, but just assume that its spectral gap
> $\lambda_1(\pmb{M}^*)-\lambda_2(\pmb{M}^*)$ is large enough.
> The randomly-generated true matrices in the experiment are not low-rank as well, which probably leads to the failure of matrix completion.
> We checked that the average of
> $\max_{i,j} \Big(| \tilde{M}_{ij}$
>
> $-M^*_{ij}| \Big)$
> was $1.0260$
> while the averages of $\max_{i,j}|M^*_{ij}|$ and $\min_{i,j}|M^*_{ij}|$ are $2.3200$ and $0.0002$, respectively, in our simulation.
> This shows that the matrix completion has not been done very precisely.
>
>
> What we can check from this experimental result is that
> if we do not have a very accurate imputation of the true matrix,
> the imputed cells introduce more noise into the inference and
> the result of sparse PCA can be even worse than that of simply using $0$ for the missing entries.
>
>
> (Continued in Response to Reviewer owug (2))
>
> Johnstone, I. M., \& Lu, A. Y. (2009). On consistency and sparsity for principal components analysis in high dimensions. Journal of the American Statistical Association, 104(486), 682-693.
>
> Ma, Z. (2013). Sparse principal component analysis and iterative thresholding. The Annals of Statistics, 41(2), 772-801.

---

### Official Review · Reviewer_jWST · 2022-07-08

**Rating:** 5
**Confidence:** 3
**Soundness:** 3 good
**Presentation:** 3 good
**Contribution:** 3 good

**Summary:**

This paper proposes a SDP relaxation of the non-convex L1-regularized sparse PCA problem under small noise and incomplete data. The resulting convex program can exactly recover the true support of the sparse leading eigenvector with high probability, under certain conditions. Experiments verify the efficacy of the proposed method.

**Questions:**

1. Given a n*d  data matrix D with missing entries (say uniformly at random), can you please provide details on how to get the d*d matrix M?

2. Since M^* is assumed to be symmetric, why not using M_ij to impute M_ji ? I think this should also be compared in the experiments.

**Limitations:**

Yes

**Strengths And Weaknesses:**

Strengths: the work provides theoretical guarantee in terms of exactly recovering the true support of the sparse leading eigenvector.

Weakness:

1. There is no comparison with any baseline methods. If I understand correctly, the method simply uses 0 for the missing entries to get matrix M. Then, the existing sparse PCA methods without the consideration of missing entries can also be applied on this matrix M, for comparison.

2.  The proposed method can only guarantee recovering the support of the sparse leading eigenvector, not the value.

---

> ### Author Response · Authors · 2022-08-02
> **Response to Reviewer jWST (2)**
>
> **Why not using $M_{ij}$ to impute $M_{ji}$?:**
>
> This has already been taken into account in the model assumption.
> Since we assume that the true matrix $\pmb{M}^*$ is symmetric,
> we naturally set the observed matrix $\pmb{M}$ to be symmetric as well.
> In other words, if $M_{ij}$ is missing, then $M_{ji}$ is also missing.
> Therefore, using $M_{ij}$ to impute $M_{ji}$ cannot be done.

---

> ### Author Response · Authors · 2022-08-02
> **Response to Reviewer jWST (1)**
>
> We thank the reviewer for the assessment and helpful comments.
>
> **There is no comparison with any baseline methods.:**
>
> To address this issue, we compared our SDP method to two different algorithms: the diagonal thresholding sparse PCA (DTSPCA) by Johnstone and Lu (2009) and the iterative thresholding sparse PCA (ITSPCA) by Ma (2013) where missing cells are treated as $0$.
> For the simulation, we set $d=50$, $s=10$, $\lambda_1(\pmb{M}^*)-\lambda_2(\pmb{M}^*) = 10$, $B=5$ and $\sigma_{normal} = 0.1$, and $20$ runs were computed.
>
> We added Figure 3 demonstrating the result in Section F of the appendix.
> From the figure, we can check that the exact recovery rate of SDP method is higher than those of DTSPCA and ITSPCA methods.
> We will include discussion on this in the final version of the main text.
>
>
> **The proposed method can only guarantee recovering the support of the sparse leading eigenvector, not the value.:**
>
> Our proposed method actually guarantee more than support recovery, and support recovery implicitly implies consistency in eigenvector estimation.
> To explain this, we note that
> under the sufficient conditions,
> the optimal solution $\hat{\pmb{X}}$ can be expressed as
> $\hat{\pmb{X}} = [\hat{\pmb{x}} \hat{\pmb{x}}^\top,~  \pmb{0} ;~ \pmb{0},~ \pmb{0}]$,
> where $\hat{\pmb{x}}$ satisfies $sign([\hat{\pmb{x}};~ \pmb{0}]) = sign(\pmb{u}_{1})$.
>
> (Please check Proposition 2 in Section B of the appendix for detail. Here we assume $J=[s]$ without loss of generality, and for $J\neq[s]$, one needs to properly interleave vector entries of $[\hat{\pmb{x}};~ \pmb{0}]$.)
> Moreover, $\hat{\pmb{x}}$ further satisfies
>
> $||\pmb{u}_{1} - [\hat{\pmb{x}} ;~ \pmb{0}] ||_2$
>
> $~~\leq \min_{j \in J} |u_{1,j}|$
>
> in our proof (see Section E.4 in the appendix.)
> That is, if we decompose our optimal solution $\hat{\pmb{X}}$ under the sufficient conditions, then we are able to obtain a vector whose elements are very close to those of
> $\pmb{u}_{1}$
>
> or $- \pmb{u}_{1}$.
> We will add a remark about this in the final version.
>
>
> **Given a $n\times d$ data matrix $\pmb{D}$ with missing entries (say uniformly at random), can you please provide details on how to get the
> $d\times d$ matrix $\pmb{M}$?:**
>
> If a data matrix $\pmb{D} \in \mathbb{R}^{n\times d}$ with missing entries is given, using $\pmb{M} = \pmb{D}^\top \pmb{D}$ is a typical approach.
> However, we note that this is a totally different setting from ours.
> We consider the case that
> a general symmetric matrix $\pmb{M}^*$ (not necessarily positive semidefinite) is observed with missing entries and the observed matrix is denoted by $\pmb{M}$.
> The case of $\pmb{M} = \pmb{D}^\top \pmb{D}$ is not our focus.
> We note that since ${\pmb{D}}^\top \pmb{D}$ is positive semidefinite and our matrices $\pmb{M}^*$ and $\pmb{M}$ are not necessarily positive semidefinite, our case is more general with respect to this aspect.
>
>
> You may wonder about the applicability of our setting over the case of $\pmb{M} = \pmb{D}^\top \pmb{D}$.
> Our setting is highly applicable especially in the fields of data privacy and federated learning.
> It is sometimes difficult for a data provider to release the covariance matrix fully due to privacy issues.
> Also, in a federated learning setting, there are cases where the central server can collect only partial information of the covariance matrix.
> Both are the cases recently receiving a great deal of attention,
> and our theorem on sparse PCA can be applied to those situations.
>
> More importantly, the matrix $\pmb{M}^*$ and $\pmb{M}$ are general symmetric matrices and not necessarily positive semidefinite,
> so our result can be applied to various data types other than covariance matrix.
> One example is an undirected random graph with missing edges.
> In this case, sparse PCA method can be used for subgraph detection problem.
> Please refer to Singh et al. (2011), for instance.
>
> Even though the case of $\pmb{M} = \pmb{D}^\top \pmb{D}$ is not our current interest, it can be considered as a future work.
> Our current analysis cannot be directly applied, because $E(\pmb{M})=E(\pmb{D}^\top \pmb{D})$ is not the true matrix ${\pmb{D}^*}^{\top} \pmb{D}^*$ anymore where
> $\pmb{D}^* = E(\pmb{D})$.
> Hence, bias correction technique may be needed in the algorithm.
> We will add this discussion in Concluding Remarks in the final version.
>
>
> (Continued in Response to Reviewer jWST (2))
>
>
> Johnstone, I. M., \& Lu, A. Y. (2009). On consistency and sparsity for principal components analysis in high dimensions. Journal of the American Statistical Association, 104(486), 682-693.
>
> Ma, Z. (2013). Sparse principal component analysis and iterative thresholding. The Annals of Statistics, 41(2), 772-801.
>
> Singh, N., Miller, B. A., Bliss, N. T., \& Wolfe, P. J. (2011, June). Anomalous subgraph detection via sparse principal component analysis. In 2011 IEEE Statistical Signal Processing Workshop (SSP) (pp. 485-488). IEEE.

---

### Official Review · Reviewer_oWbf · 2022-07-12

**Rating:** 5
**Confidence:** 4
**Soundness:** 2 fair
**Presentation:** 2 fair
**Contribution:** 2 fair

**Summary:**

## Summary
The authors study the semi-definite relaxation of the sparse principal component analysis (PCA) under missingness and establish the sufficient conditions for support recovery of the first principal component.

## Novelty
Support recovery of the semi-definite relaxation of the sparse PCA with complete observations was established by Lei and Vu (2015), this work extends the analysis to incomplete observations.

## Overview of Comments

I really appreciate the clarity and the level of detail, but describing all these in the main paper devoids it of context and seems like a listing of constants and their mathematical form. Due to this, the paper is somewhat unreadable and does not (because of space) address the key difference in analysis and specific challenges it had to overcome (and how) over the previous works. This description is key for the readers to gauge the merits. For instance, the details of how the analysis differs from Lei and Vu (2015) under the missingness is unclear.

**Questions:**

## Comments

1. Applicability of the model in case of missing observations. The paper considers decomposition of a symmetric matrix with noisy and potentially missing observations. Arguably in real-world for a data matrix $Y$, this symmetric matrix was derived by performing $M = Y'Y$. But decomposition of $M$ may be ill-posed when columns of data matrix $Y$ have a few observed entries [See Park & Zhao 2019]. It seems that the theoretical results must impose some additional conditions on $Y$ to avoid such a situation.
   - 1(a) The paper discusses these indirectly in section 3, but implications for original data $Y$ is not discussed. Although it seems these must depend on the incoherence and the sampling probability, these issues must be directly addressed.
   - 1(b) The bound on p would be more interpretable if directly expressed, than an upper bound on $\sqrt(1-p/p)$
   - 1(c) the various notions of incoherence employed must be introduced in the model description since these are inherent assumptions about the matrix $M$.


2. In Lei and Vu (2015) the exact support recovery relies on Sparse Principal Subspace (SPS) condition, which includes 1) spectral gap and 2) $supp(diag(\Pi)) = J$, where $J = Union_{j=1}^{k} supp(u_j)$, where $u_j$ are the eigenvectors. In essence, $J$ is not the support of the principal eigenvector as the proposed work but a union of all eigenvectors.

   - 2 (a) Since the proposed model does not impose sparsity of other principal components and they can be dense, it is unclear if support recovery is even possible if other eigenvectors are dense. What are the additional conditions required even for the complete observation setting.

   - 2(b) This also seems like a special case of sparse PCA when only the learning principal vector is sparse.

3. Writing -- majority of the write-up is dedicated to definitions of constants and results. I appreciate this level of details, but these are best left to the appendix. In the current form these are extremely distracting tending to unreadable.

   - 3(a) Since the papers relies on Landau notation, these need to further simplified. This is also important for the experiments where the paper claims a that the results are "tight" but these relationships are not plotted for reference.

   - 3(b) What are the differences in analysis over Lei and Vu (2015)? What do the incoherence conditions imply in practice?


**Strengths And Weaknesses:**

## Strengths

* Precise about the analysis

## Weaknesses

* While the paper is precise about the analysis, it focuses on listing these results and quantities in the main paper, distracting the readers from the key points of the paper. This makes the paper difficult to read. This also makes it difficult to parse the main result and its implications.
* The paper does not highlight the key differences in analysis and challenges in establishing the results for this case over the prior art.
* Applicability of the model for real-world applications in unclear (see detailed comments/questions).

---

> ### Author Response · Authors · 2022-08-02
> **Response to Reviewer oWbf (3)**
>
> **Majority of the write-up is dedicated to definitions of constants and results. These are best left to the appendix.:**
>
> Most of the main text is to describe our key results clearly, and hard to be simplified or deferred to the appendix, in our opinion.
> However, we acknowledge that
> the paragraphs of 'Conditions on $p$ (ratio of missing data)' (lines 186-195) in page 6 can be condensed.
> We suggest to simplify the paragraphs as follow:
>
>
> > **Conditions on $p$ (ratio of missing data)**
> > For simplicity, suppose that $\bar{\lambda}(\pmb{M}^*_{J,J}) = O(s)$ and $s=O(\log d)$.
> > We consider two extreme cases where the coherence parameters are maximized and minimized.
> > We discuss the bound of the sample complexity in each case.
> >
> > $\bullet$ *The best scenario where the bound of the sample complexity is the lowest*:
> >
> > Suppose that $\mu_1 = o(\frac{1}{\log s})$ and
> > $\mu_2 = \mu_3 = 1$ (note that when $\mu_0 = o\big(\frac{1}{\sqrt{s}\log s}\big)$, $\mu_1$ is upper bounded by $o\big(\frac{1}{\log  s}\big)$.)
> > Then the condition (6) can be written as:
> > $
> > p = \frac{1}{1+\epsilon_1}, ~~
> > \epsilon_1 = o\bigg(\min \Big(
> > \frac{1}{\log s},~~
> > \frac{1}{sd||\pmb{M}^*_{J^c,J}||^2_{\max}}
> > \Big)
> > \bigg).
> > $
> > If $||\pmb{M}^*_{J^c,J}||_{\max} = O(\sqrt{\frac{\log s}{sd}})$, then $p = \omega\Big(\frac{1}{(\log s)^{-1} + 1}\Big)$.
> >
> > $\bullet$ *The worst scenario where the bound of the sample complexity is the highest*:
> > Suppose that
> > $\mu_1 = \frac{1}{\sqrt{s}}$, $\mu_2 = \frac{1}{\sqrt{s(d-s)}}$ and $\mu_3 = \frac{1}{d-s}$.
> > In this case, the condition (6) can be written as:
> > $p = \frac{1}{1+\epsilon_2}, ~~ \epsilon_2 = o\bigg(\min \Big(\frac{\log s}{s}, \frac{1}{ (sd||\pmb{M}^*_{J^c,J}||_{\max})^2 },$
> >
> > $ \frac{1}{ (d ||\pmb{M}^*_{J^c,J^c}||_{\max})^2 } \Big) \bigg).$
> >
> > For instance, if $||\pmb{M}^*_{J^c,J}||_{\max} = \Theta(\frac{1}{s\sqrt{d}})$ and
> >
> > $||\pmb{M}^*_{J^c,J^c}||_{\max} = \Theta(\frac{1}{\sqrt{d}})$,
> > then $p = \omega\Big(\frac{1}{d^{-1} + 1}\Big)$.
>
> Here, the definitions of $\epsilon_1$ and $\epsilon_2$ are different from those of the original text.
>
> We hope this change enhances the readability.

---

> > ### Comment · Reviewer_oWbf · 2022-08-08
> > **Response to the authors**
> >
> > Thank you for the response. I would like to clarify if the authors intend on submitting a revised manuscript with the proposed changes. If yes, it would be great if they can highlight the changes in a different text color. Thanks!

---

> > > ### Author Response · Authors · 2022-08-09
> > > **Response to Reviewer oWbf**
> > >
> > > Our supplementary material has been updated, aiming to address all the comments from reviewers on additional experiments, but not the main text.
> > > We will follow the due process, and in case the paper gets accepted, we will definitely revise the main text to incorporate suggestions from reviewers, for which we will use the extra page.
> > > For now, we have addressed several points during the rebuttal phase.
> > >
> > > In particular, we have added Sections F and G, and Figures 3, 4 and 5 in the supplementary material. We have marked this part in red.

---

> ### Author Response · Authors · 2022-08-02
> **Response to Reviewer oWbf (2)**
>
> **About Comment 2:**
>
> Our work focuses on the case that the leading eigenvector is sparse and the others can be dense. Our sparsistency assumption is that 1) the gap between first and second largest eigenvalues of the true matrix $\pmb{M}^*$ is relatively large and 2) its leading eigenvector is sparse, where the support of the leading eigenvector $supp(\pmb{u}_1)$ is denoted by $J$. That is, $J$ is not the union of supports of multiple eigenvectors here.
>
> When multiple leading eigenvectors are sparse and the true support $J$ is the union of the supports of those vectors,
> it is not trivial to find meaningful sufficient conditions for exact recovery of $J$ under missingness (for instance, as mentioned above, simply applying concentration inequalities in the proof of Lei and Vu (2015) results in too strict conditions on sample complexity and data.)
>
> We note that there have been a great deal of works focused on the case that only the leading eigenvector is sparse (d’Aspremont et al. (2004), Amini and Wainwright (2009), Journee et al. (2011), Berthet and Rigollet (2013) and Yuan and Zhang (2013)).
> This implies that sparse PCA in our current setting is still widely applicable and leads to great successes in applications such as genome wide association studies and image compression.
>
> **About Comment 1(b):**
>
> We will change the expression $\sqrt{\frac{1-p}{p}} = o(f)$ to $p = \frac{1}{1+o(f^2)}$ for better interpretation.
>
> **About Comment 1(c):**
>
> We will move Definition 1 of coherence parameters to Section 2.2.
>
> **About Comment 3(a):**
>
> The term "tight" used in experiment 1 of Section 4.1 means that the exact recovery rate is solely determined by the rescaled parameter of $p$ derived from the theorem (this definition is mentioned in line 237.)
> We can numerically check this by plotting the exact recovery rate versus the rescaled parameter, not $p$.
> If all of the curves of different settings line up with one another, then we can say that our theoretical prediction and empirical behavior agree with each other.
> We showed this in Figure 1.
> This kind of technique was also used in Wainwright (2009).
>
> We note that "tight" does not mean an agreement between necessary and sufficient conditions here, which probably caused confusion.
> We will add a remark for this in the final version.
>
> (Continued in Response to Reviewer oWbf (3))
>
> d'Aspremont, A., Ghaoui, L., Jordan, M., \& Lanckriet, G. (2004). A direct formulation for sparse PCA using semidefinite programming. Advances in neural information processing systems, 17.
>
> Amini, A. A., \& Wainwright, M. J. (2008, July). High-dimensional analysis of semidefinite relaxations for sparse principal components. In 2008 IEEE international symposium on information theory (pp. 2454-2458). IEEE.
>
> Journée, M., Nesterov, Y., Richtárik, P., \& Sepulchre, R. (2010). Generalized power method for sparse principal component analysis. Journal of Machine Learning Research, 11(2).
>
> Berthet, Q., \& Rigollet, P. (2013). Optimal detection of sparse principal components in high dimension. The Annals of Statistics, 41(4), 1780-1815.
>
> Yuan, X. T., \& Zhang, T. (2013). Truncated Power Method for Sparse Eigenvalue Problems. Journal of Machine Learning Research, 14(4).
>
> Wainwright, M. J. (2009). Sharp thresholds for High-Dimensional and noisy sparsity recovery using $\ell _ {1} $-Constrained Quadratic Programming (Lasso). IEEE transactions on information theory, 55(5), 2183-2202.

---

> ### Author Response · Authors · 2022-08-02
> **Response to Reviewer oWbf (1)**
>
> We thank the reviewer for the constructive feedback and suggestions.
>
> **The details of how the analysis differs from Lei and Vu (2015) under the missingness is unclear.:**
>
> Given space limitations, we mostly focus on describing results in detail, e.g., what conditions on data and sample complexity are required.
> We agree that a discussion on the differences of our analysis over prior works (e.g., Lei and Vu (2015)) adds more value to our work. Please find such discussion in the below paragraph, and it will be included in the final version.
> However, we would like to respectfully say that the absence of this discussion at the present stage cannot be the reason for rejection.
>
> To deal with missingness, we utilized concentration inequalities such as matrix Bernstein inequality and Chebyshev’s inequality, which is one key technical difference from the analysis of Lei and Vu (2015).
> Beyond this,
> under the primal-dual witness framework, (while Lei and Vu (2015) employed implicit constraints in the optimization problem,)
> we made all the constraints explicit and directly derived the sufficient conditions without using any auxiliary lemmas in Lei and Vu (2015).
> In other words, our proof road map is completely different from that of Lei and Vu (2015).
> Our proof successfully avoids strict conditions on sample complexity and data matrix.
> In detail, simply applying concentration inequalities to the approach of Lei and Vu (2015) results in the following sufficient condition on the submatrix $\pmb{M}^*_{J^c, J^c}$:
>
> $C(||\pmb{M}^*_{J^c, J^c}||_{\max}\log d$
>
> $~~ + \sqrt{p(1-p)} ||\pmb{M}^*_{J^c, J^c}||_{2,\infty} \sqrt{\log d}) \leq \rho$
>
> for some positive constant $C$.
> This condition is hard to be satisfied when the matrix dimension $d$ is large, since it requires the matrix norm of $\pmb{M}^*_{J^c, J^c}$ and the probability $p$ to be small and close to $1$, respectively.
> On the other hand, with our approach, the condition on $\pmb{M}^*_{J^c, J^c}$ can be much relaxed as follows:
>
> $C'(||\pmb{M}^*_{J^c, J^c}||_{\max}\log d$
>
> $~~+ \sqrt{p(1-p)}||\pmb{M}^*_{J^c, J^c}||_{2,\infty} \sqrt{\log d}) \leq \rho s$
>
> where $s$ is the size of support $J$ and $C'$ is some positive constant. When $s = \Omega({\log d})$, there are no longer stringent requirements on $\pmb{M}^*_{J^c, J^c}$ and $p$ with a high dimension $d$.
>
>
> **About Comment 1(a):**
>
> A data matrix $\pmb{Y}^* \in \mathbb{R}^{n\times d}$ being observed with missing entries (denote the observed matrix by $\pmb{Y}$) and the matrix $\pmb{M} = \pmb{Y}^\top \pmb{Y}$ being used in the optimization is a totally different setting from ours.
> We consider the case that a general symmetric matrix $\pmb{M}^*$ (which is not necessarily positive semidefinite) is incompletely observed and the observed matrix denoted by $\pmb{M}$ is used in the optimization.
> The case of $\pmb{M} = \pmb{Y}^\top \pmb{Y}$ is not our focus, so of course the conditions on $\pmb{Y}^*$ are not addressed in our work.
> We note that since ${\pmb{Y}^*}^\top \pmb{Y}^*$ and ${\pmb{Y}}^\top \pmb{Y}$ are positive semidefinite and our matrices $\pmb{M}^*$ and $\pmb{M}$ are not necessarily positive semidefinite, our case is more general with respect to this aspect.
>
> Our setting is highly applicable especially in the fields of data privacy and federated learning.
> It is sometimes difficult for a data provider to release the covariance matrix fully due to privacy issues.
> Also, in a federated learning setting, there are cases where the central server can collect only partial information of the covariance matrix.
> Both are the cases recently receiving a great deal of attention,
> and our theorem on sparse PCA can be applied to those situations.
>
> More importantly, the matrix $\pmb{M}^*$ and $\pmb{M}$ are general symmetric matrices and not necessarily positive semidefinite,
> so our result can be applied to various data types other than covariance matrix.
> One example is an undirected random graph with missing edges.
> In this case, sparse PCA method can be used for subgraph detection problem.
> Please refer to Singh et al. (2011), for instance.
>
> Even though the case of $\pmb{M} = \pmb{Y}^\top \pmb{Y}$ is not our current interest, it can be considered as a future work.
> Our current analysis cannot be directly applied, because $E(\pmb{M})=E(\pmb{Y}^\top \pmb{Y})$ is not the true matrix ${\pmb{Y}^*}^{\top} \pmb{Y}^*$ anymore where
> $\pmb{Y}^* = E(\pmb{Y})$.
> Hence, bias correction technique may be needed in the algorithm.
> We will add this discussion in Concluding Remarks in the final version.
>
> (Continued in Response to Reviewer oWbf (2))
>
> Singh, N., Miller, B. A., Bliss, N. T., \& Wolfe, P. J. (2011, June). Anomalous subgraph detection via sparse principal component analysis. In 2011 IEEE Statistical Signal Processing Workshop (SSP) (pp. 485-488). IEEE.

---

### Meta-Review · Area_Chair_NX5i · 2022-09-02

**Recommendation:** Accept
**Confidence:** Less certain

**Metareview:**

The paper studies the sparse PCA problem — finding sparse direction of large variance. It studies the natural convex relaxation, in which we lift the outer product xx^T to a matrix X, drop the rank constraint, and apply L1 regularization. In contrast to much existing work, here, the covariance matrix is partially observed. The proposal is to apply the convex relaxation, with the covariance matrix M^* replaced by a censored and noisy version M. The paper assumes that M^* has a sparse lead eigenvector, and ask when this approach correctly recovers the support of that vector.

Reviewers found the paper to be technically solid, with novel results on sparse PCA with missing data. The main concerns center around the presentation: the style is dense with definitions and conditions, perhaps to the detriment of the reader’s insight. Arguably, with a better chosen set of sufficient conditions would lead to a more readable collection of results, leaving more space to convey what the results mean and how they were achieved.


**Award:**

No

---

### Decision · Program_Chairs · 2022-09-14

Accept